

**Assessment of land use impact on hydraulic threshold conditions for gully head cut**
**initiation**
Aliakbar Nazari Samani[1], Qiuwen Chen[2,3,*], Shahram Khalighi[1,3], Robert James Wasson[1]
1. Natural Resources Faculty, University of Tehran, Karaj, Iran
2. CEER Nanjing Hydraulics Research Institute, Nanjing, 210023, China
3. RCEES Chinese Academy of Sciences, Beijing, 100085, China
**\*** Correspondence to: Tel./Fax: +86 10 62849326, E-mail: qwchen@nhri.cn



**Abstract:** Gully erosion is a geomorphic threshold phenomenon controlled by different
environmental factors as well as human activities. In this research, we examined the effect of
land use on hydraulic flow and the consequent head cut initiation for similar soil conditions
using an experimental plot of 15m*0.4m. Results indicated that boundary shear stresses $\tau_{cr}$
for gully initiation in rangeland, dry farming and abandoned land are 192, 43 and 174
dyne/cm$^2$, respectively, due to the differences in surface vegetation cover. Moreover, the
turbulence of flow and soil response to an increase in water depth showed complicated
behavior, which could be attributed to the effect of surface micro relief features and land use
impacts. Compared to dry farming, the short vegetation cover in the rangeland decreased the
effect of ground cover on flow regime. Even after seven years of abandonment, the response
of agricultural land to increasing shear stress was similar to that of dry farming, which
indicated the low resilience and high erosional susceptibility of soil in dry land
environments. The main explanation for dramatic (3-4 fold) variations of $\tau_{cr}$ was the
vegetation cover and soil surface conditions. In fact, the remarkable decrease of $\tau_{cr}$ in dry
farming was related to the effect of tillage practice on soil susceptibility and aggregate
strength. The findings indicated that a critical shear stress of 35 dyne/cm$^2$ used in some
physically based models for erosion prediction is not appropriate for estimating gully
erosion. In addition, the duration of land abandonment has a crucial influence on soil
erodibility that has been less considered in erosion models.
**Keywords**: erosion; gully head cut; shear stress; threshold; land use




## 1 Introduction

Concentrated water erosion phenomena are usually classified according to three categories, namely rill, stream and gully. Researchers use different criteria to separate and characterize rills, streams and gullies (which are also separated into ephemeral and permanent gully forms). Hauge (1977) first used a critical cross-sectional area of 929 cm$^2$ and Brice (1966) introduced a minimum width and depth of 0.3 m and 0.5 m respectively as a criterion to distinguish rill from gully (Imenson and Kwaad 1980). Although the transition from rill to gully erosion is a continuum process, Torri et al. (1987) and Bryon and Slattery (1992) went a step further and suggested a hydraulic concept for rill and gully formation.

Although gullies are responsible for most sediment yield in many catchments in a wide range of environmental conditions, such as described by Nazari Samani et al. (2011) in Iran, Wasson et al. (1996) and Poesen et al. (2003) in Europe, and Li et al. (2003) in China, many soil erosion models have focused mainly on sheet and rill erosions, neglecting soil loss by gullies at the catchment scale (Poesen et al. 2003).

Gully erosion is clearly a geomorphic threshold phenomenon. Gullies can develop only if concentrated (overland) flow intensity during a rainfall event exceeds a threshold value and flow surpasses the soil resistance. This force of flow is often expressed in terms of the boundary flow shear stress ($\tau_s = \gamma ds$ with $\gamma$=density of runoff water, kg/m$^3$; d=depth of flow, m; and s=slope of the soil surface gradient, m/m). The threshold force required to cause channel incision into the soil surface in the concentrated flow zone is termed the 'critical flow shear stress' ($\tau_{cr}$). In addition, detailed investigation into the relationship between the geomorphic threshold and shear stress revealed that upslope drainage area and slope gradient are linked to critical shear stress (Begin and Schumm, 1979). The combination of hydraulic and geomorphic thresholds produced the $\Gamma_{cr} = (c\gamma)A^{rf}S$ relationship, where $\Gamma_{cr}$ is the critical shear stress indicator, A is the drainage area (ha), S is the slope gradient (m/m), rf and c are





experimental coefficients (Vandaele et al., 1996). Some researchers have shown the effect of
land use on the geomorphic threshold (e.g Vandekcheknov et al., 2000; Poesen et al., 2003;
Nazari Samani et al., 2009), but less attention has been paid to the hydraulic conditions of
head cut initiation. In fact a key question is: how large should $\tau_{cr}$ be in order to initiate a gully
head cut? More recently, the threshold has been defined quantitatively using two criteria,
namely shear stress and stream power. Govers (1985) was the first to conceptualize a shear
threshold velocity for rill erosion initiation by conducting a flume experiment in loamy soils.
To date, several experiments have been conducted to investigate the hydraulic threshold
condition for head cut initiation: Prosser et al. (1995, 2000) in the grassland near San
Francisco, Nachtergaele and Poesen (2002) in the Belgian loess belt, and Adelpour (2004) in
loamy-sands in Iran. The different results obtained by these researchers indicate that more
field-based tests are needed to better determine the effect of land use on the threshold
condition for head cut initiation. In addition, in some physically based erosion models (such
as WEPP, CREAMS, PRORILL and EGEM), shear stress is a key parameter and the value
used in WEPP is 3.5 Pa. Therefore, it is very important to study the critical shear stress for
different ground surface conditions so as to understand the effective factors and develop a
comprehensive erosion model. The main objective of this study is therefore to determine how
land use factors affect flow conditions (status, type and threshold shear stress), and
consequently the initiation of head cut erosion.
**2 Materials and methods**
**2.1 Experiment design**
The experiments were conducted in the Samal area located in the Dareh-Kore watershed of
Boushehr province in south of Iran. The region has a typical arid to semi-arid climate with an
average annual temperature of 14 ℃ and an average annual rainfall of 200 mm. The main
lithological formations include the Miocene Fars Group (Aghajari, Mishigan; consisting of




marl, shale, marly and shaly limestone) in the uplands and Quaternary alluvium (consisting of
gravels, sands, silt and clay) in the piedmont plain. Gully erosion and badland formation are
two highly destructive processes impacting on the hilly and lowland areas, and are common
on the Quaternary formations with slope gradients of less than 20%.
The flume experiments were conducted using an erosion plot that was 15 m long and 0.4 m
wide and 0.5 m high, designed to create non-uniform flow resistance. The ground surface
cover of the soil was not disturbed. For each experiment, the parameters of hydraulic flow
were measured over the 9 m reach in the middle of the flume (Fig. 1 and 2). Three land uses,
dry farming, rangeland and abandoned areas, were chosen. In addition, in order to prevent the
effects of spatial variation of soil properties, all tests were conducted at a site consisting of
three land uses. The distance between test locations was about 200 m. The soil attributes
according to the land use are presented in Table 1, which show that no significant difference
was found in the soil attributes, although a small variation in the samples could be seen in the
Ca, organic matter and Na. However, slope could not be held constant. The maximum soil
surface slope was in the rangeland (5.9%), while the dry farming land had the least surface
slope (0.13%). Therefore, in order to determine the effect of land slope, the shear stress index
($\tau = \gamma.R.S$) was used. This index considers both discharge and energy characteristics, which
are explained further in the following section. The characteristics of the land cover in the
experimental sites were as follows:
a. Rangelands: No surface gravel and uniform cover of lichens and mosses (Fig. 1), with

grasses (5%) of *St.cap, St.ar*, and low litter (1%).

b. Dry farming: Ground cover of annual grasses (*Ho. Sp.*; *Br. tec.*), forbs (40%) (*Ch. Ab.,*
*As. Sp.,*) and residuals of stalks from previous years and no surface gravel. In contrast
to rangelands, the canopy cover of the dry farming land is much greater because of
agriculture operations and low slope as well as establishment of weeds.



c. Abandoned areas: This land had been relinquished for 7 years. Vegetation cover of 50%

includes annual grasses (*Agi. sp*, *Ma. Sp*, *Fu. sp*, *Br. tec.*) and forbs, low gravel cover

(1%) and litter (3%).

[Fig. 1 is here]

[Fig. 2 is here]

[Table 1 is here]

**2.2 Experimental operation, measurement and parameter calculation**
The flume's sidewalls were beaten into the soil and sealed with plaster, cement and soil to
prevent leakage and incursions by animals. To determine the slope of the longitudinal profile
with high precision, ground surveying was performed using a theodolite camera, leveling rod
and measuring tape. After setting up the water supply equipment including a water tank,
stilling basin and Parshal flume at both ends of the plot, the experiment was started with low
discharge (0.75 liter per second) then increased to high discharge so that the head cut could
be observed. For every experiment, the flow parameters including discharge, depth of flow
(by a steel ruler) and sediment samples (at the end of the flume) were measured directly,
while the water surface velocity was determined by liquid dye tracers (injected once). The
following relations were used to calculate the hydraulic characteristics of flow.
Mean flow velocity:  $U = \dfrac{Q}{A}$                                      (1)
Reynolds number:  $Re = \dfrac{U.d}{\upsilon}$                               (2)
Froude number: $F = \dfrac{U}{\sqrt{gy}}$                                     (3)
Shear stress of flow:  $\tau = \gamma.R.S$                                   (4)
The soil detachment rate:  $D_r = \dfrac{C_v.Q.t}{6}$                        (5)
where Q is Discharge (m³/s), A is cross section area of flow (m²); U is mean flow velocity



(m/s); d is flow depth (m); $\upsilon$: kinematic viscosity ($\upsilon = 0.01 \text{cm}^2/\text{s}$); F is Froude number; g is
gravitational acceleration (m/s²); y is mean flow depth (m); γ is specific gravity (ρg); S is
water surface slope; R is hydraulic radius (m); $C_v$ is sediment weight concentration (kg/m³); t
is run time (s).
To assess the soil detachment rate based on threshold shear stress, the following relation was
established (Foster 1982; Nachtergaele et al., 2002):
$D_r = Kc(\tau - \tau_{cr})^B$                                       (6)
where $D_r$ is the detachment capacity of flow (kg m⁻² s⁻¹); Kc represents the soil erodibility to
concentrated flow (S/m); τ is mean shear stress; $\tau_{cr}$ is critical shear stress; B is an empirical
coefficient usually equal to 1. Thus, equation 6 can be written in linear mode (Eq. 7).
$D_r = K_c\tau + b$                                       (7)
Comparison of equations 6 and 7 indicates that the intercept *b* can be related to critical shear
stress via $\tau_{cr} = \dfrac{-b}{K_c}$. Consequently, by plotting $D_r$ versus shear stress and fitting with a linear
line, the slope of the fitted line is equal to Kc.
The initiation of a gully was obtained by visual and photo monitoring of the flume surface for
each run. A small ditch or hole is sufficient to permit head cut initiation. Therefore an
incision of 3*3 cm-size was adopted as an index of head cut initiation. A rural well dug near
the site was selected as the water resource for supplying the water through a petrol driven
pump. A retention pond with an overflow pipe was established in the upward end of flume. A
total of four, seven and five runs were conducted on the rangeland, dry farming and
abandoned land respectively to reach the mentioned threshold of head cut initiation.
**3 Results**
**3.1 Effect of land use on type of flow**
As mentioned previously, the regime and type of flow were quantified by using Reynolds (Re)





and Froude numbers (Fr) respectively (Table 2). Generally, in all the experiments, flow status
was turbulent (Re > 2000). In contrast to other land use, in rangelands, because of the short
grass cover and smooth lichen surface, its effect on the Reynolds number was very low (low
surface roughness). However, in dry farming land, due to the high vegetation cover, the
Reynolds number was greatly affected during low discharge. In fact, the land covers in dry
farming and abandoned lands increased the surface roughness and indirectly caused the
decrease of Re by decreasing the flow velocity. But as discharge and consequently flow depth
increased and flow overtopped and submerged the canopy, mean velocity increased while Re
increased to a remarkable value of 25,000. It is noticeable that during mean discharge (4 lit/s
in Table 2), Re in rangelands was lower than in both dry farming and abandoned lands,
leading to an increase in flow energy. The main reasons for this increased turbulence could be
the micro topography on the soil surface in abandoned areas and dry farming lands in
comparison to rangeland.

[Table 2 is here]

The Froude number (Table 2) varied from 0.05 to 5.1. Head cut initiation was observed with
Fr=1.61 (Q = 9.2 lit/s); Fr= 0.1 (Q = 8.2 lit/s) and Fr= 0.6 (Q = 4.3 lit/s) for rangeland, dry
farming and abandoned land respectively. In other words, as the soil surface was disturbed,
such as by tillage operation, a flow with less energy was sufficient to initiate a head cut. A
head cut could be initiated both under and above critical flow conditions. However, the
discharge needed to create enough energy for incision in rangeland was more than was
required for dry farming and abandoned lands.
Figure 3 depicts an example of surface profiles for various discharge experiments in
abandoned land. It can be seen that with low discharge, due to the impact of the roughness of
vegetation cover and micro topography, the profile of the water surface (run 1 in Fig 3) is
similar to that of the ground profile. But as the flow depth increases, the water surface





becomes smoother. In fact, vegetation cover can influence both the flow characteristics (e.g.
flow resistance, roughness and flow depth) and the hydraulic attributes such as the rating
equation of flow depth with discharge and boundary layer depth. Therefore it is postulated
that an alteration of the vertical velocity profile causes turbulent flow as stems and branches
are overtopped.

[Fig. 3 is here]

**3.2 Impact of land use on the threshold shear stress for surface erosion**
The results of the relationship between the detachment rate ($D_r$) and the shear stress are
shown in Figures 4, 5 and 6. We preferred to use dyne/cm$^2$ as the shear stress unit because of
the small values obtained in units of Pa (1Pa=10 dyne/cm$^2$). As can be observed, in all cases,
significant relationships (P=0.05) between Dr and shear stress were observed. The threshold
shear stress for each land use was calculated based on the slopes and intercepts shown in
Figures 4, 5 and 6. These values are 83, 11 and 74 dyne/cm$^2$ for rangelands, dry farming lands
and abandoned areas respectively. Moreover, soil erodibility for concentrated overland flow
($K_c$) was obtained for rangeland (0.0038) and dry farming (0.1912). It is notable that the
resistance of soil to concentrated flow in rangelands is more than 50 times that in dry farming
land.

[Fig. 4 is here]

[Fig. 5 is here]

[Fig. 6 is here]

**3.3 Effect of land use type on gully initiation threshold**
The numbers of head cuts corresponding to mean shear stress for each experiment were listed
in Table 3. The critical shear stress for head cut initiation was 174 dyne/cm$^2$ in rangeland, 35
dyne/cm$^2$ in dry farming, and 153 dyne/cm$^2$ in abandoned land. The 3-4 fold difference
between the calculated critical shear stresses in the three studied land uses could be linked to





the soil surface condition. Although the vegetation cover of rangeland was less than that of
dry farming, the biological crust of lichens and mosses made the soil very resistant to
detachment. In fact, the presence of biological crusts on the surface of the soil in the
rangeland increased the surface soil resistance several-fold (Table 2, Fig. 4 and 5). Table 3
demonstrates that the number of head cuts increased with shear stress. For example, from run
3 to run 5 in abandoned land, the number of head cuts increased more than two-fold while the
average shear stress increased just 1.3 times.

[Table 3 is here]

From Table 3 and Fig. 6, it can be found that the relationship between head cuts and shear
stress of abandoned land was similar to the dry farming lands, although the critical shear
stress for head cut initiation of abandoned land (153 dyne/cm$^2$) was close to that of rangeland
(174 dyne/cm$^2$).
**4 Discussion**
From the study, it was found that for the rangeland, which had a natural cover, soil
detachment and head cut initiation occurred under a sub-critical flow regime. The calculated
Froude number was between 0.65 and 1.10, which was consistent with other findings that a
Froude number between 0.5 and 2.8 was the threshold value for incision (Knapen et al., 2006;
Adelpour, 2004; Prosser et al., 1995). The main reason for the low Froude number in dry
farming areas was the high vegetation cover and roughness. However, soil disturbances
caused by previous tillage operations decreased the strength of aggregates dramatically;
consequently, flow detached and entrained soil particles more easily, which led to the
creation of head cuts.
In dry farming and abandoned areas, due to the high vegetation cover and low depth flow, a
sub-critical regime was observed. But as the flow depth increased, the overtopping of
branches and stalks diminished the impact of vegetation cover. Despite sub-critical flow in



dry farming and abandoned lands, the detachment rate was more than twice that of rangeland.
This could be mainly attributed to the decrease in aggregate resistance produced by tillage
operations (Knapen *et al.*, 2007). Furthermore, it seemed that the impacts of vegetation cover
changes depended on the roughness effect. In arid and semi-arid climates where vegetation
cover was very low, any change in land cover could dramatically affect the roughness, and
therefore the soil detachment and erosion (Léonard and Richard, 2004). The relationship
between average shear stress, contributory catchment area and slope proposed by Begin and
Schumm (1979) showed the role of a geomorphic threshold on shear stress. Based on the
relationship, it is seen that as $\tau_{cr}$ increases, upslope area and slope gradient must increase in
order to initiate a gully. Nazari et al. (2009) reported that in this study area, when land use
changed from rangeland to dry farming land, the areas susceptible to gullying increased by a
factor of two, from 6% to 12% of the total area. Therefore, land use changes not only affected
soil stability but also decreased the geomorphic threshold, causing more areas prone to
gullying.
In addition, the impacts of tillage operations on the aggregate attributes such as degree of
consolidation, soil weathering, dry and wetness, can affect the $K_r$ parameter (Franti *et al.*,
1985; King *et al.*, 1995). This study showed that land use change could increase soil
erodibility ($K_r$) more than 50 times and decrease boundary shear stress about 6 fold. This
meant that the effect of land use change on $K_r$ was more significant than on $\tau_{cr}$. Similar results
have been reported by other researchers (Nagchtargle and Poeson, 2002; Knapen *et al.*, 2007),
who found that using the conventional K in the USLE cannot reflect the spatial variations of
erodibility in a landscape scale. With the same soil attributes, both the vegetation cover and
the micro relief of the ground surface are the main factors determining the spatial variation of
detachment and sedimentation along the flume (Bergsma and Farshand, 2004), preventing the
establishment of a stable and uniform erosion pattern. To assess and model erosion over a



landscape, a simple sediment transport equation does not give a precise result regarding
detachment and sedimentation (Morgan, 2005; Adelpour, 2004). Therefore, the adoption of a
large range of $K_r$ values is essential to improve physically based erosion models.
It was noticeable that $K_r$ of the abandoned land and rangeland were similar in low run-off
depth (run 1 and 2 in Table 2). However, $K_r$ of the abandoned land in high run-off depth (run
3 in Table 2) was different from that of the rangeland, while it was similar to that of the dry
farming land. Such behavior indicated that for a given soil, a change of land use affected the
run-off erosion process for several years. The value of $\tau_{cr}$ for head cut initiation on the
rangeland is five times higher than that in dry farming land, implying that high surface and
subsurface (10 cm) aggregate resistance in the rangeland was probably a result of the
biological crust.
The mean $\tau_{cr}$ for the whole dataset of this research was 134 dyne/cm$^2$, which was lower than
the global average value of 150 dyne/cm$^2$ (Knapen et al., 2007). The main reason for this
difference could be the discrepancy of ground features and the use of a sandy loam soil. The
relationships between the numbers of observed head cuts and shear stress in the abandoned
area and rangeland were the same when $\tau_{cr}<140$ dyne/cm$^2$. However, in abandoned land, as
the $\tau_{cr}$ increased, the observed number of head cuts increased by a factor of three (Table 2).
This was because land use not only affected the resistance of the surface soil but also affected
the resistance of the sub-soil. After seven years of abandonment, the erodibility of sub-soil
had not changed significantly. Even though no tillage operations had been conducted on the
abandoned land for seven years, the sub-soil had not or even could not return to its original
condition and level of resistance.
**5 Conclusions**
Experimental results of detachment and head cut initiation indicated that critical shear stress
($\tau_c$), soil resistance to concentrated flow ($K_c$) and head cut initiation were dependent on land



use and soil surface conditions. Critical shear stress has been the most widely used parameter
for physically-based models such as WEPP, EPP, EUROSEM and CREAMS. It was
concluded from this study that most physically based models should use a wider range of
both $K_r$ and $\tau_{cr}$ values. In other words, the use of a single value of $\tau_{cr}$=35 dyne/ cm$^2$ as the
boundary shear stress cannot accurately represent the threshold condition for gully initiation.
In addition, the duration of farming land abandonment should be taken into consideration in
order to obtain a realistic value for $K_r$. Further experiments are needed to quantify the effects
of land use and soil attributes on gully initiation so as to obtain a generally applicable model
for gully erosion.
**Acknowledgements**
This research was supported by the Center of Excellence for Sustainable Watershed
Management of the Tehran University and National Nature Science Foundation of China
(51425902). We appreciate Dr. Catherine Rice, PhD from Purdue University, USA for
proofreading the English.





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



**Tables**
Table 1: Soil attributes of three land uses selected for experiments

| Land use | Texture | Silt (%) | Ec (ds/m) | OC (%) | Lime (%) | Na (meq/lit) | Ca (meq/lit) | SAR | pH | Cl (meq/lit) | Ground slope (%) |
|---|---|---|---|---|---|---|---|---|---|---|---|
| Rangeland | Sandy loam | 8 | 3.74 | 0.44 | 23.30 | 33 | 18.4 | 7.8 | 7.3 | 15.6 | 5.9 |
| Dry farming | Sandy loam | 5.5 | 3.44 | 0.85 | 23.75 | 34 | 15 | 8.1 | 7.3 | 16.4 | 0.13 |
| Abandoned | Sandy loam | 5 | 3.34 | 0.50 | 21.25 | 29.5 | 14 | 7 | 7.3 | 14.7 | 4.4 |



Table 2: Results related to the status and type of the flow in different experiments

|  | Discharge (l/s) | Mean flow depth (mm) | Fr number | Re number |
|---|---|---|---|---|
|  | 2 | 13 | 1.11 | 5037 |
|  | 3.9 | 17 | 1.46 | 9860 |
| Rangeland | 6.37 | 23 | 1.46 | 16190 |
|  | 9.2 | 28 | 1.61 | 24178 |
|  | 0.75 | 50 | 0.05 | 1834 |
|  | 1.21 | 62 | 0.06 | 3065 |
|  | 3.5 | 95 | 0.1 | 8817 |
| Dry farming | 4.1 | 101 | 0.1 | 10361 |
|  | 5.7 | 129 | 0.1 | 14391 |
|  | 8.2 | 165 | 0.1 | 20757 |
|  | 10 | 170 | 0.11 | 24969 |
|  | 1.5 | 15 | 0.69 | 3913 |
|  | 2.9 | 22 | 0.72 | 7299 |
| Abandoned | 4.3 | 31 | 0.64 | 10937 |
|  | 5 | 34 | 0.65 | 12756 |
|  | 7.2 | 41 | 0.67 | 18289 |



Table 3 Shear stress for different runs with observed head cuts for each land use
(1Pa=10 dyne/cm$^2$).

| Land use | Run | Mean shear stress along the flume (dyne/cm$^2$) | Number of head cuts | Critical shear stress for head cut initiation (dyne/cm$^2$) |
|---|---|---|---|---|
| Rangelands | 1 | 70 | | |
| | 2 | 106 | | 174 |
| | 3 | 146 | 1 | |
| | 4 | 178 | 2 | |
| Dry farming land | 1 | 5 | | |
| | 2 | 9 | | |
| | 3 | 15 | | 35 |
| | 4 | 19 | | |
| | 5 | 34 | 2 | |
| | 6 | 42 | 5 | |
| Abandoned areas | 1 | 78 | | |
| | 2 | 115 | | |
| | 3 | 161 | 3 | 153 |
| | 4 | 178 | 5 | |
| | 5 | 217 | 8 | |




**Figures**


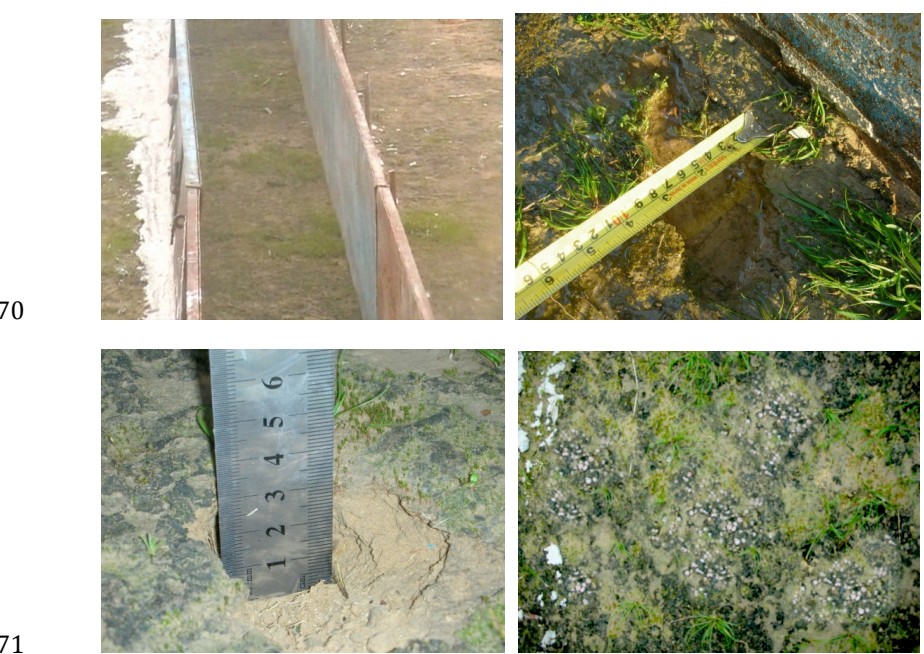

Fig. 1 Flume and ground vegetation (grass and lichen) in the rangeland plot with
ground slope of 6% (top left). Ground surface measurement (top right) and head cut
features with step height of 3 cm (bottom left). Lichens and mosses on the ground
surface of rangeland soil (bottom right).






Fig. 2 Schematic of experimental flume showing the mid-section used to measure
flow depth and ground elevation.





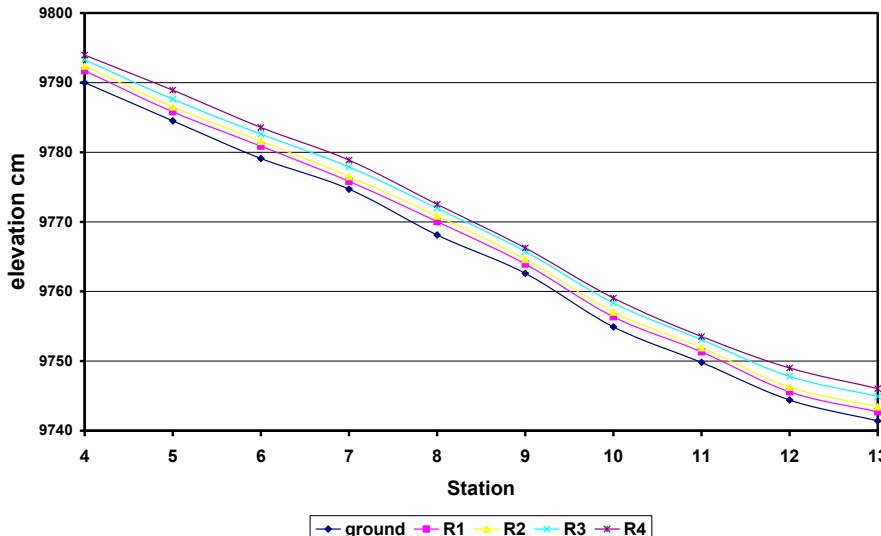


Fig. 3 Profile of water surface and bed of plot 1 in the undisturbed condition. Presence
of non-uniform vegetation cover had led to decreased roughness coefficient and
increased flow velocity. Consequently, flow depth decreased between points 7 and 10.
The points were based on the Fig. 1 scheme. The space between two points was 1m
and the slope gradient was about 4.5%.





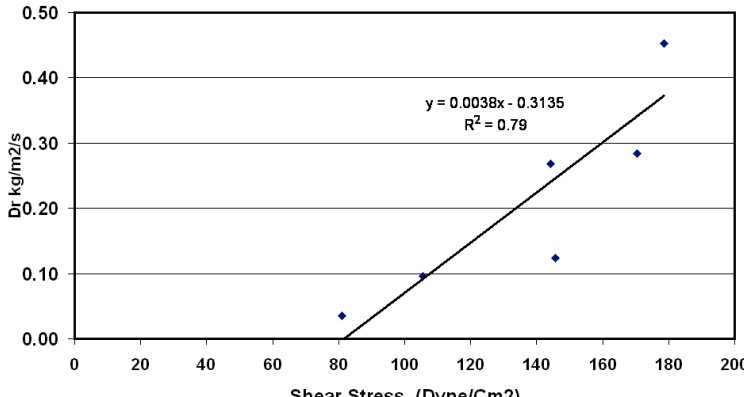


Fig. 4 The relationship between shear stress ($\tau$) and detachment rate in the rangeland





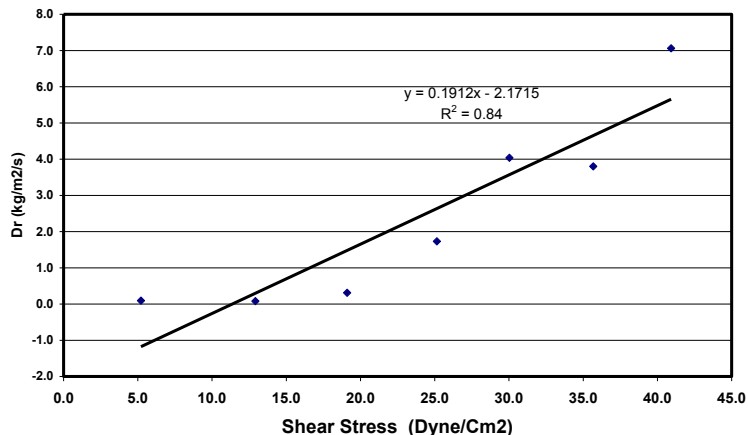


Fig. 5 Relationship between shear stress ($\tau$) and detachment rate in dry farming land





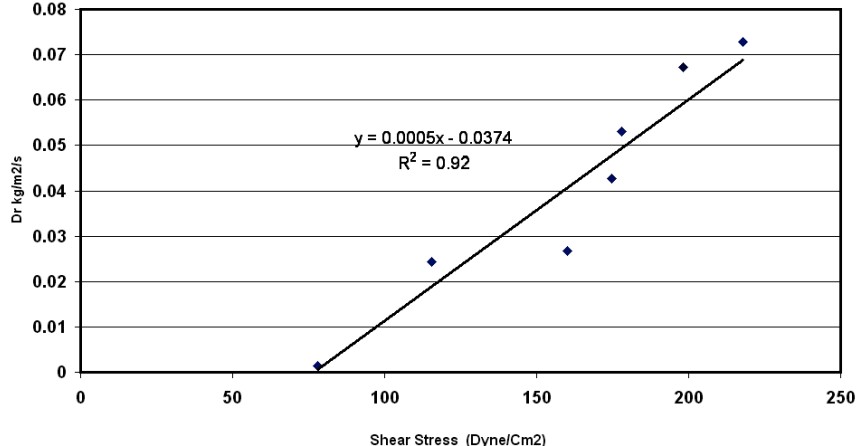


Fig. 6. Relationship between shear stress (τ) and detachment rate in abandoned land