# Peer review of "Assessment of land use impact on hydraulic threshold conditions for gully head cut"

_Hydrology and Earth System Sciences, 2015_

## Referee Comment (RC1) · Aliakbar Nazari Samani et al. · 10 Mar 2016

This is an interesting paper to assess the impacts of land use on type of flow, the threshold shear stress for surface erosion and gully initiation threshold. The overall presentation is well structured and clear. The conclusions are useful for the development of physically based erosion models. However, this paper leaves out some details that should be properly explained. Specific comments and some technical corrections are discussed in the following.

Specific comments

1) Lines 114-116 says that "the experiment was started with low discharge (0.75 liter per second) then increased to high discharge so that the head cut could be observed". First, information about how the discharge increased from low to high discharge gradually is better to be given here. Further, in my view, the three experiments are better to have same runs and same discharge in each run so that the experiments can be more comparable and scientific. However, this study designed different runs and discharges for the three experiment land (Table 2). Please explain why design in this way. 2) Line 144-146 states that "A total of four, seven and five runs were conducted on the rangeland, dry farming and abandoned land respectively to reach the mentioned threshold of head cut initiation". But there are 6 runs shown in Table 3 for the dry farming land. Please check. 3) Lines 154-156 stated "the land covers in dry farming and abandoned lands increased the surface roughness and indirectly caused the decrease of Re by decreasing the flow velocity." So my question is that the flow velocity used to calculate Re in this study is the measured values or calculated by equation (1)? 4) Few references cited in manuscript are from last five years. Please read more literatures in recent years and modify the introduction.

Technical corrections

1) The mathematical symbols in the formulae and in the paper should keep consistent. The "U" in formula (1) is in italics while is in non-italics in other places (say line 125). The "b" in equation (7) and on line 137 is inconsistent. 2) The Froude number is expressed using "F" on line 126 while using "Fr" on line 150. Please check the whole paper and unify their form. 3) Experiments of the three kinds lands are better to be separated with a horizontal line in Table 2 and Table 3 so that readers can distinguish which discharges or runs corresponding to which land use type more easily. 4) please check the sentence on lines 168-170 "However, the discharge needed to create enough energy for incision in rangeland was more than was required for dry farming and abandoned lands." 5 The Kc in equation (7), c is subscript or not? Please check.

---

## Referee Comment (RC2) · Anonymous Referee #2 · 16 May 2016

The manuscript entitled "Assessment of land use impact on hydraulic threshold conditions for gully head cut initiation" presents experimental results regarding gully head initiation under different land covers. The major findings basically explain the dependency of critical shear stress to land cover conditions. Although the experimental setup and results are valuable, I believe the discussion is relatively weak. In several cases, the findings are already known as mentioned by the authors. Here, I provide some comments which can improve the quality of this manuscript; In several occasions the effect of upslope area on head initiation has been mentioned. However, the experimental results do not include any information in this regard. The upslope area and slope are two well studied channel initiation thresholds. Since the gully head resembles channel heads in drainage network, I believe studying upslope area of gully heads would be valuable. Did the gully head create a connected network (similar to channel heads that form a channel network)? If yes, I think studying the characteristics (the density, branching behavior and spatial distribution) of the resulted network would be even more interesting than just focusing on the gully heads. My main concern about the experimental setup is the initial condition of soil in terms of moisture content. Was this considered in the experiment and how was its effects isolated? The discussion, mainly attributes the land cover to the erosional susceptibility of soil, however, I believe, the infiltration is also important here. Land cover affects the infiltration (as simply quantified in SCS-CN) and therefore impacts the erosional force (volume and velocity of overland flow) through the mass balance. Specific comments; Line 89- "which indicates no significant difference in the soil attributes, .." Line 111- To determine longitudinal slope with high precision, Table 2- I suggest represent the results visually in some figures. Line 169- was more the one for dry farming . . . Figure 3- The decreasing trend of depth is hard to observe in this figure. It is better to plot the depth rather than elevation. Line 182- There is a typo Line 185- It is better to report P in each figure. Line 187- I think 83 is not correct: b=-Tcr*Kc Tcr = b/Kc=0.3136/0.00038=825 Line 197- How the head imitation shear stress is calculated here? Section 3.3- It is better to represent Mean shear stress versus Number of heads in a figure and then discuss the relationship. Line 278: Kr or Kc? Line 288- delete "the English".

―――――――――――――――――

---

## Author Comment (AC1) · 22 May 2016

Specific comments 1. Lines 114-116 says that "the experiment was started with low discharge (0.75 liter per second) then increased to high discharge so that the head cut could be observed". First, information about how the discharge increased from low to high discharge gradually is better to be given here. Further, in my view, the three experiments are better to have same runs and same discharge in each run so that the experiments can be more comparable and scientific. However, this study designed different runs and discharges for the three experiment land (Table 2). Please explain why design in this way.

– The main objecive of this study was to determine of threshold condition for head cut

initiation. This threshold is different under different land use, therefore we had to follow the land condition. In fact a given discharge in one land use could not be applied to aother due to slope and land cover variation. Moreover, the situation of land in three treatments was not the same and consequently we have to pursue different runes. We increased the discharge gradually, and after each run the surface of flume was monitored. In lines 124-133, we explain this issue in details.

2. Line 144-146 states that "A total of four, seven and five runs were conducted on the rangeland, dry farming and abandoned land respectively to reach the mentioned threshold of head cut initiation". But there are 6 runs shown in Table 3 for the dry farming land. Please check.

– The Authors would like to appreciate such realistic and accurate view-point from the reviewers about the data reporting. In fact, the table 3 is related to the head cut initiation. As stated in the text, seven runs were conducted on the dry farming. The sixth and seventh runs were very similar; therefore we decided to neglect the sixth run for the head cut initiation. In addition, after the 5th run, the flow exceeded the threshold condition. We have added these information in the revised manuscript. Also, we consider this point and completed the data in the table 2.

3. Lines 154-156 stated "the land covers in dry farming and abandoned lands increased the surface roughness and indirectly caused the decrease of Re by decreasing the flow velocity." So my question is that the flow velocity used to calculate Re in this study is the measured values or calculated by equation (1)?

– The velocity was calculated. To check the results, surface velocity was also measured by liquid dye tracers. In the revised MS, we focused on the Fr number and the stream power analysis instead of flow type and Reynolds number. Please refer to Section 3.1 page 12 of the revised manuscript.

4. Few references cited in manuscript are from last five years. Please read more literatures in recent years and modify the introduction.

– We tried to find some new relevant references (Zhang et al., 2014; Tekwa et al., 2015). We added some recent literature and correspondingly we carried out some new analysis for stream power and threshold shear stress calculation. There are a lot of literatures about gully erosion, and most of them concentrate on the factors affecting gully processes. However, few studies are available that test the hydraulics condition and gully initiation under field and undisturbed soil condition. We believe that real field data and experiments are needed for understanding the gully erosion process, which is the major sediment sources.

Technical corrections 1. The mathematical symbols in the formulae and in the paper should keep consistent. The "U" in formula (1) is in italics while is in non-italics in other places (say line 125). The "b" in equation (7) and on line 137 is inconsistent.

– The symbols and formula were carefully checked and revised.

2. The Froude number is expressed using "F" on line 126 while using "Fr" on line 150. Please check the whole paper and unify their form.

– Many thank indeed to the reviewers for your precise review. It has been fixed and the Fr is correct.

3. Experiments of the three kinds lands are better to be separated with a horizontal line in Table 2 and Table 3 so that readers can distinguish which discharges or runs corresponding to which land use type more easily.

– Thanks indeed for your great suggestion. The Table was carefully revised fixed (page 21)

4. Please check the sentence on lines 168-170 "However, the discharge needed to create enough energy for incision in rangeland was more than was required for dry farming and abandoned lands."

– Thanks indeed for your suggestion. In the revised MS, the entire paragraph and this specific sentence have been rewritten (page 9, lines 195-203).

5. The Kc in equation (7), c is subscript or not? Please check.

– Thanks indeed for your suggestion. c is subscript. I had been revised.

---

## Author Comment (AC2) · 22 May 2016

Specific comments:

1. The upslope area and slope are two well studied channel initiation thresholds. Since the gully head resembles channel heads in drainage network, I believe studying upslope area of gully heads would be valuable.

– Thanks indeed for your positive comment. The geomorphic threshold (area- slope) are two major parameters. In the line 62 -68 of the revised manuscript, we tried to add the relationship between Are-Slope and shear stress. This subject has been fully considered for this region in the previous study (see reference below). In this research,

the objective is to use the experimental flume to investigate the effects of land use on threshold condition for gully initiation.

Nazari Samani A, Ahmadi H, Jafari M, Guy B, Ghoddousi J, and Malekian A (2009) Geomorphic threshold conditions for gully erosion in southwestern Iran (Boushehr-Samal watershed). Journal of Asian Earth Sciences 35: 180–189.

2. Did the gully head create a connected network (similar to channel heads that form a channel network)? If yes, I think studying the characteristics (the density, branching behavior and spatial distribution) of the resulted network would be even more interesting than just focusing on the gully heads.

– Thanks indeed for the valuable comment. We fully this point. The initial stage of gullying process is head cut generation with a small size. The nature of our research was based on a experimental flume (15 m), which is located in the area close to gullies network. But none of treated field was the active connected gully network, because we tried to investigate the head cut initiation at the flume scale, where the initial forms and small head cuts were emphasized. The density, branching behavior and spatial distribution can be very important for watershed scale study of gully erosion modelling, which is our next step.

3. The discussion mainly attributes the land cover to the erosional susceptibility of soil, however, I believe, the infiltration is also important here. Land cover affects the infiltration (as simply quantified in SCS-CN) and therefore impacts the erosional force (volume and velocity of overland flow) through the mass balance.

– We are very gteaful to the ctitical comment. The infiltration and soil hydrology are exactly two main components for erosion modelling. In the revised manuscript, the discussion was developed to consider this point using the Q*Fr criteria (Line 256-264). In addition, to create the similar condition between experiments, all of the tests were stablished under the saturated condition with very-very low infiltration. In the new revised manuscript (line 127-135), the method was described in detail. To create a steady flow,

we had to make the infiltration conditions reach to saturation. By this the differences between inflow and out flow was negligible. In the new Fig. 2, such condition was described.

Technical corrections

1.Line 89- "which indicates no signifi̧cant difference in the soil attributes,. Line 111- To determine longitudinal slope with high precision, Table 2- I suggest represent the results visually in some fi̧gures. Line169-was more the one for dry farming ... Figure3- The decreasing trend of depth is hard to observe in this fi̧gure. It is better to plot the depth rather than elevation

– The manuscript has been thoroughly revised. For example the section 3.1 and figure 3 were removed and a new section was added. The table 2 was replaced by three other figures for better visualization. Also in the revised manuscript the mentioned sentences were checked and rewritten.

2.Line 182- There is a typo Line 185- It is better to report P in each fi̧gure.

– It has been done.

3. Line 187- I think 83 is not correct: b=-Tcr*Kc Tcr = b/Kc=0.3136/0.00038=825

– Many thanks indeed for your correction. We carefully checked it and found it is a typo mistake. In fact for rangeland the Tcr = b/Kc value based on the figure 4 is 0.3135/0.0038 = 82.5. The slope of fitted line is 0.0038 not 0.00038

4. Line 197- How the head imitation shear stress is calculated here?

– Thanks for your question to better calrify the manuscript. In table 3, the critical shear stress for each head cut was calculated by the shear stress equation $\gamma RS$, where the S and R variables were obtained from measured data according to Figure 2 (in the mid-section of the flume).

5. Section 3.3- It is better to represent Mean shear stress versus Number of heads in

a figure and then discuss the relationship.

– In the discussion section of revised MS, the response of land uses to increase of shear stress have been taken into account (Line 429-443). Because of the different land uses' data sets, the plot is not uniform. In addition, plotting the data is not enough as in each land use we have two or three data sets. If we plot whole data within onw graph, it can not show a uniform scatter pattern (see below graph). However, if the respected reviewer believes that this graph is more informative than the table, we can replace it in the final version.

6. Line 278: Kr or Kc? Line 288- delete "the English".

– Thanks indeed for your correction. It is Kc, and it has been revised.

[Figure]

**Fig. 1.** Relationship between number of head cut and average shear stress